# Localization Approach for Underwater Sensors in the Magnetic Silencing Facility Based on Magnetic Field Gradients

**DOI:** 10.3390/s22166017

**Published:** 2022-08-12

**Authors:** Guohua Zhou, Yufen Wang, Kena Wu, Hanming Wang

**Affiliations:** School of Electrical Engineering, Naval University of Engineering, Wuhan 430033, China

**Keywords:** localization, underwater sensors, magnetic field gradients, magnetic silencing facility

## Abstract

Localization of the underwater magnetic sensor arrays plays a pivotal role in the magnetic silencing facility. A localization approach is proposed for underwater sensors based on the optimization of magnetic field gradients in the inverse problem of localization. In the localization system, a solenoid coil carrying direct current serves as the magnetic source. By measuring the magnetic field generated by the magnetic source in different positions, an objective function is established. The position vector of the sensor is determined by a novel multi-swarm particle swarm optimization with dynamic learning strategy. Without the optimization of the magnetic source’s positions, the sensors’ positions, especially in the *z*-axis direction, struggle to meet the requested localization. A strategy is proposed to optimize the positions of the magnetic source based on magnetic field gradients in the three directions of *x*, *y* and *z* axes. Compared with the former method, the model experiments show that the proposed method could achieve a 10 cm location error for the position type 2 sensor and meet the request of localization.

## 1. Introduction

Marine vessels, such as submarines and ships, consist of ferromagnetic material which can disturb the Earth’s magnetic field [1,2]. The significance of a marine vessel’s magnetic field has been proverbial since Germany blockaded Britain with magnetic naval mines, causing great losses to the British Navy in World War II. Magnetic silencing facilities (MSF) are utilized to reduce the risk of damaging submarines and ships from mines and the magnetic airborne detection [3]. Its key elements are magnetic sensor arrays on the seafloor, whose efficacious performance depends on the accuracy of the sensors installed and the accuracy of locating the sensors after installation [4]. According to the technique of installing sensors, the magnitude of the position deviation vector in the three-axis direction is no more than 30 cm, which could bring magnetic field errors more than 100 nT and have repercussions for the magnetic assessment of marine vessels. The existence of the position deviations directly reduces the accuracy of the ship’s measured magnetic field and affects the assessment of the ship’s magnetic protection capability.

For the magnetic silencing facilities, underwater sensors are installed at a depth of 15~20 m from the pier plane, where a high-sensitivity GPS hardly works in the sea as it is incompatible with water [5,6,7]. In addition, due to seaweed organisms, accumulated sediment and murky water, there are complex conditions in seafloor environments. Therefore, it is difficult to provide precision with the localization methods which use both acoustic and optical signals [8]. Comparatively, using a magnetic signal is more useful in a shallow maritime environment with the advantage of energy efficiency and low cost. Many methods have explored how to locate underwater sensors [9,10]. Callmer explored a method to locate underwater sensors by using an extended Kalman Filter and a vessel with known static magnetic signature, but it is hard to obtain highly accurate data for the ragged static magnetic signature [11]. Yang proposed a method to simplify the 3D position problem into 2D problem by using depth sensors, then it was optimized [12]. Zhang investigated a method which equipped the underwater sensors with auxiliary coils [13]. Both of these methods require the installation of auxiliary components and would thus be expensive. Cerro proposed a magnetic localization approach by means of TMR triaxial sensors on a finite domain, whose results place the measurement accuracy lower than 1 cm when the localization range is a cube of 30 cm. However, the uniaxial (*z* axis) magnetic sensors are still widely applied in the magnetic silencing facility [14]. Yu proposed a method which used an improved non-dominated sorting genetic algorithm and linear multi-metering method to locate the triaxial sensors by building a magnetic source with known positions and magnetic moment [15].

In this paper, a localization approach based on magnetic field gradients (MFG) is proposed to determine the positions of the magnetic source in the inverse problem of localization. Firstly, a suitable length is set as the grid spacing and meshes the measurement plane. The magnetic field of the region centered on the sensor is calculated when the magnetic source is placed at every grid node, then the average gradients of the magnetic field in the three-axis directions are calculated. By using the average gradients, the positions of the magnetic source are determined and the overdetermined system of the equation is established. Through optimization algorithms, the sensor positions are determined.

The arrangement of this paper is organized as follows. In Section 2, we discuss how to simplify the magnetic source and present the localization approach based on the magnetic field gradients. In Section 3, the localization approach applied in the typical facility is analyzed and verified by numerical experiments. In Section 4, the experiments are carried out. Finally, we draw a conclusion in Section 5.

## 2. Localization Model and Approach

### 2.1. Magnetic Dipole Source

Figure 1 shows section view of the magnetic silencing facility, where the sensor arrays are arranged at the bottom of the sea and the solenoid coil carrying direct current (magnetic source) is located on the plane of the pier. One of the vertices of the plane on the pier is set as the original point *o* of the global coordinate system *o*-*xyz* and the center of the solenoid coil is set as the original point *o*′ of the solenoid coil coordinate system *o*′*-x*′*y*′*z*′.

In the solenoid coil system, the magnetic flux density ***B*** = [*B_x_*, *B_y_*, *B_z_*]^T^ is generated by the solenoid coil, which is a high-order nonlinear function of the sensor position *P* = [*p_x_*, *p_y_*, *p_z_*]^T^ with three variables. Additionally, the magnetic flux density ***B*** = [*B_x_*, *B_y_*, *B_z_*]^T^ in the sensor position *P* = [*p_x_*, *p_y_*, *p_z_*]^T^ created by the solenoid coil could be exported as follows [16]:(1)$$\begin{array}{l} B(P,o^{\prime })\\ =\frac{\mu_{0}N_{w}I}{2\pi \sqrt{{\left(\sqrt{p_{x}^{2}+p_{y}^{2}}+R\right)}^{2}}+p_{z}^{2}}\\ \times \{\frac{p_{x}p_{z}}{p_{x}^{2}+p_{y}^{2}}\left[\frac{r^{2}+R^{2}}{\sqrt{{\left(\sqrt{p_{x}^{2}+p_{y}^{2}}-R\right)}^{2}}+p_{z}^{2}}E(k)-K(k)\right]{e^{\prime }}_{x}\\ +\frac{p_{x}p_{z}}{p_{x}^{2}+p_{y}^{2}}\left[\frac{r^{2}+R^{2}}{\sqrt{{\left(\sqrt{p_{x}^{2}+p_{y}^{2}}-R\right)}^{2}}+p_{z}^{2}}E(k)-K(k)\right]{e^{\prime }}_{y}\\ \left[\frac{R^{2}-r^{2}}{\sqrt{{\left(\sqrt{p_{x}^{2}+p_{y}^{2}}-R\right)}^{2}}+p_{z}^{2}}E(k)+K(k)\right]{e^{\prime }}_{z}\} \end{array}$$
where *μ*_0_ is the magnetic permeability of vacuum (*μ*_0_ = 4π × 10^−7^), *R* is the coil radius, *I* is the direct current loaded, *N_w_* is the number of solenoid coil turns, *r* is the distance between the sensor position *P* and the solenoid coil center *o*′, *e*′*_x_*, *e*′*_y_* and *e*′*_z_* are the base vectors of the coil coordinate system, and
(2)$$\begin{array}{l} k^{2}=4R\sqrt{p_{x}^{2}+p_{y}^{2}}/\left[{\left(\sqrt{p_{x}^{2}+p_{y}^{2}}+R\right)}^{2}+p_{z}^{2}\right]\\ K(k)=\int_{0}^{2\pi }d\alpha /\sqrt{1-k^{2}\sin^{2}\alpha }\\ E(k)=\int_{0}^{2\pi }\sqrt{1-k^{2}\sin^{2}\alpha d\alpha } \end{array}$$

During the calculation and iteration, the calculation of calculus such as Equation (1) in each circulation is very time-consuming. If the distance *r* between the center of the coil and the sensor position is large enough in comparison with the coil radius *R* [17], the solenoid coil could be simplified as a magnetic dipole source within the allowable range of errors. The uniaxial (*z* axis) magnetic sensors are still widely applied in the military facilities. Therefore, the vertical component of the magnetic flux density *B_z_* is used to determine the localization approach in this paper. Additionally, the vertical component of magnetic flux density *B_z_* in the sensor position *P* = [*p_x_*, *p_y_*, *p_z_*]^T^ by the magnetic dipole source is exported as follows [18]:(3)$$B_{z}(P,o^{\prime })=\frac{\mu_{0}\left[3p_{x}p_{z}m_{x}+3p_{y}p_{z}m_{y}+(2p_{z}{}^{2}-p_{x}{}^{2}-p_{y}{}^{2})m_{z}\right]}{4\pi {\left(p_{x}{}^{2}+p_{y}{}^{2}+p_{z}{}^{2}\right)}^{\frac{5}{2}}}$$
where ***m*** = [*m_x_*, *m_y_*, *m_z_*] is the magnetic dipole moment. To facilitate the implementation of the localization approach, the size of the magnitude of the magnetic dipole moment *m* is optimized. Maximizing *m* within the allowable range of the magnetic dipole errors is preferred for localization. The uniaxial (*z* axis) magnetic sensors are still widely applied in the military facilities. Therefore, the vertical component of magnetic flux density *B_z_* is used for the localization approach in this paper. By changing the positions of the magnetic source, underwater sensors measure the magnetic field with different values and the system of equations is established.

### 2.2. Localization Approach

The magnetic dipole moment ***m*** could be obtained by measurement and inversion, when the sensor position *P* = [*p_x_*, *p_y_*, *p_z_*]^T^ is determined accurately as the installed position. the vertical component of the magnetic flux density *B_z_* computed by Equation (3) is the same as the measured magnetic data *B_zm_* by the sensor. Therefore, localization uses the measured magnetic data *B_zm_* to solve the inverse problem:(4)$$\left\{ \begin{array}{l} \mathrm{Minimize}\;f(P)\\ f(P)=||B_{z}-B_{zm}{\left|\right|}_{2} \end{array}\right.$$

The essence of localization is minimizing the difference in the computed magnetic data *B_z_* and the measured magnetic data *B_zm_* to determine the sensor position *P* = [*p_x_*, *p_y_*, *p_z_*]^T^. However, considering the fact that Equation (3) is a function of the sensor position *P* = [*p_x_*, *p_y_*, *p_z_*]^T^ with three variables, a minimum of three equations are required to solve the sensor position and form the well-posed system of equations. Generally, the overdetermined system of equations is established to improve the accuracy of localization. The systems are defined as:(5)$$\left\{ \begin{array}{l} \mathrm{Minimize}\;f\;(P)=(f_{1}(P),f_{2}(P),f_{3}(P),\ldots ,f_{i}(P))\\ \;\;\;\;\;\;\;\;\;\;\;\;\;\;\;\;\;\;f_{1}(P)=||B_{z}{}^{1}-B_{zm}{}^{1}{\left|\right|}_{2}\\ \;\;\;\;\;\;\;\;\;\;\;\;\;\;\;\;\;\;f_{2}(P)=||B_{z}{}^{2}-B_{zm}{}^{2}{\left|\right|}_{2}\\ \;\;\;\;\;\;\;\;\;\;\;\;\;\;\;\;\;\;\ldots \\ \;\;\;\;\;\;\;\;\;\;\;\;\;\;\;\;\;\;f_{i}(P)=||B_{z}{}^{i}-B_{zm}{}^{i}{\left|\right|}_{2}\left(i\geq 3\right) \end{array}\right.$$
where ***B_z_*** = [*B_z_*^1^, *B_z_*^2^, *B_z_^3^*,…, *B_z_^i^*] is the calculated magnetic data and ***B_zm_*** = [*B_zm_*^1^, *B_zm_*^2^, *B_zm_*^3^,…, *B_zm_^i^*] is the measured magnetic data when the coil is placed in different positions. Generally, a linear multi-metering method (LMM) from a latest research study [15] can be applied, as illustrated in Figure 2. The coil moves discontinuously from *C*_1_ to *C*_k_ along a straight line and a set of magnetic field data is measured simultaneously.

In order to acquire more accurate positions of sensors with minimal cost, an approach based on magnetic field gradients (MFG) is proposed to determine the positions of the magnetic source, as shown in Figure 2. Firstly, the plane of the pier is divided into fixed-size grids with an appropriate grid spacing. Considering the fact that the magnitude of the position deviation vector in the three-axis direction is no more than 30 cm, the average magnetic field gradients in the *x*-axis, *y*-axis and *z*-axis directions of the region 60 × 60 × 60 cm centered on the sensor is calculated, respectively, when the magnetic source is placed at every grid node. Simultaneously, the coil coordinates are recorded. Then, the average gradients of magnetic field in the three directions are sorted from large to small, respectively as ***G_x_*** = [*G_x_*^1^, *G_x_*^2^, *G_x_*^3^, …] (*G_x_*^1^ is the largest value in the sorting in *x*-axis direction and so on), ***G_y_*** = [*G_y_*^1^, *G_y_*^2^, *G_y_*^3^, …] and ***G_z_*** = [*G_z_*^1^, *G_z_*^2^, *G_z_*^3^, …]. The positions of the magnetic source are determined according to the gradients and the request of accuracy. When only three measurements can reach the request of localization, the coil coordinates corresponding to the average magnetic field gradients ***G*****_1_** = [*G_x_*^1^, *G_y_*^1^, *G_z_*^1^] are determined, respectively. If six measurements are found, it corresponds to ***G*_2_** = [*G_x_*^1^, *G_y_*^1^, *G_z_*^1^, *G_x_*^2^, *G_y_*^2^, *G_z_*^2^], and so on.

Finally, the system of equations could be solved by optimization algorithms, such as a novel multi-swarm particle swarm optimization with dynamic learning strategy [19]. The selection of optimization algorithms has two traits. One is including avoiding falling into local optimality and another is reducing the time of optimization. Then, the accurate positions of sensors could be obtained.

## 3. Numerical Experiments

In this section, a numerical simulation experiment platform is built according to the typical magnetic silencing facility shown in Figure 2, which shows a two-dimensional cross section view of the typical magnetic silencing facility, where the sensors with a depth of 15 m can be divided into two position types. The first and second position types are 4 m and 10 m away from the pier, respectively.

In addition, the first and second position types are called type 1 and 2, respectively. In part A, the localization approach proposed is compared to the linear multi-metering method. Then, the grid spacings and the number of magnetic source positions are analyzed. Numerical experiments with different position deviation vectors are carried out in part B.

### 3.1. Analysis of Different Grid Spacings and Number of Magnetic Source Positions

In the experiments, considering the magnetic dipole equivalent errors, the magnetic dipole moment ***m*** is set as 31,415 A·m^2^. The size of the pier’s plane is 10 × 60 m and its z coordinate is 0 m. Simultaneously, considering the actual situation of the facility with a “drive-in” system, the measurement noise and resolution of underwater sensors are 10 nT and 1 nT, respectively. Firstly, the linear multi-metering method (LMM) is applied to the localization of type 2 for comparison with the localization approach proposed in this paper, as shown in Figure 3. The grid spacing is set as 2 m, the position deviation vector is set as (30, 30, 30) with the unit of centimeter. The absolute error and its relative error are defined as follows:(6)$$\begin{array}{l} d_{r}=\sqrt{{\left(p_{x}-p_{x_{0}}\right)}^{2}+{\left(p_{y}-p_{y_{0}}\right)}^{2}+{\left(p_{z}-p_{z_{0}}\right)}^{2}}\\ d_{e}=d_{r}/\sqrt{p_{x_{0}}{}^{2}+p_{y_{0}}{}^{2}+p_{z_{0}}{}^{2}}\times 100\% \end{array}$$
where *P* (*p_x_*, *p_y_*, *p_z_*) is the position to be determined and *P_0_* (*p_x_*_0_, *p_y_*_0_, *p_z_*_0_) is its installed position of the sensor.

Figure 3 shows the location results of MFG and LMM. Clearly, as the number increases, all errors are gradually reduced. Compared with LMM, the location results using MFG are more accurate and could achieve up to a 10 cm positioning error.

For the location process clear, Figure 4a–c shows when the number of the magnetic source positions is set as 9, three groups of coordinates of the magnetic source are determined by magnetic field gradients in the three directions of *x*, *y* and *z* axes, respectively. In addition, the three groups of coordinates are marked with *x* coordinates, *y* coordinates and magnetic field gradients.

Figure 4d shows the coordinates of the magnetic source of MFG and LMM, wherein the six coordinate points are coincident and only the remaining three are different, but the location accuracy varies.

In order to analyze the impact of the grid spacings and the number of the magnetic source’s positions, the grid spacings are set as 1 m, 2 m, 3 m, 4 m and 5 m and the number of the magnetic source’s positions is set as 3 to 36 with an interval of 3, successively. Table 1 and Table 2 show the average absolute errors and maximum absolute errors, respectively. Both kinds of errors decrease with added measurement points and descend into a saturation phase in the end. Overall, with the grid spacings from 1 to 5 m, the location errors show small differences. Type 1 and type 2 could achieve up to 5 cm and 10 cm location errors, respectively. Compared with type 2, the results of type 1 show more accurate data for a closer distance.

### 3.2. Numerical Experiments

Different deviation vectors are set to test the localization approach proposed in this paper. Due to the limitation of space and instruments in the laboratory, the grid spacing is set as 2 m and the number of the magnetic source’s positions is set as 12. In addition, five deviation vectors of 30 cm are set and the numerical experiments of localization are carried out.

Table 3 shows results of localization of two sensors by MFG and LMM. Compared with LMM, the results using MFG are more accurate, especially in the maximum absolute errors (*d_r_*-max). Both of the average absolute errors have a weak difference, near 1 cm, but their difference in the maximum absolute errors could reach about 5 cm.

In the approach proposed, for the type 1 sensor, the average absolute error and the maximum absolute error fluctuate around 3.2 cm and 9.5 cm, respectively. For the type 2 sensor, the average absolute error and the maximum absolute error fluctuate around 6.6 cm and 19.3 cm, respectively. Facing different deviation vectors, the localization approach is accurate.

## 4. Model Experiment

To validate the practical feasibility of the localization approach proposed in this paper, the physical scale model experiments are carried out as shown in Figure 5.

Considering the laboratory limitations, the scale of the model is set as 1:8.5 and only the type 2 sensor is located. In addition, the number of the magnetic source positions is set as 12. There were five position deviation states set and the error amplitude was set as 3 cm. The coil radius R is 0.1 m, which is less than one-fifth of the distance between the center of the coil and the sensor position. The resolution and the range of the magnetic sensors were 1 nT and ±60 μT, respectively.

Table 4 shows the result of the experiments. The absolute position error using LMM is about 1.2 cm and its relative error is 0.6 %. The absolute position error using MFG is about 0.87 cm and its relative error is 0.4 %. Compared with LMM, the results show that the proposed approach is more accurate, which is consistent with the results of the numerical experiments outlined above and validates the practical feasibility of the localization approach. When zooming in proportionally, its absolute position error is 10.2 cm, which is near 10 cm and thus meets the magnetic field measurement requirements of ferromagnetic ships.

## 5. Conclusions

A localization approach based on the optimization of magnetic field gradients has been proposed to determine the sensors’ positions in the inverse problem of localization. The plane of the pier is fixed-size grids with a fit spacing. Then, by calculating the gradients of the vertical component of the magnetic field in the region centered on the sensor, the positions of the measurement points are determined according to the gradients in the *x*-axis, *y*-axis and *z*-axis directions, respectively. The position vector of the sensor is determined by a novel multi-swarm particle swarm optimization with dynamic learning strategy. The experimental results show that the proposed strategy based on the optimization of magnetic field gradients (MFG) is effective. Its high accuracy of position type 1 and 2 has also been validated. Compared to LMM, the model results show that the absolute position error of position type 2 could reach 10 cm in a practical use scenario.

In this paper, three groups of coordinates of the magnetic source are determined by magnetic field gradients in the three directions of the *x*, *y* and *z* axes to build the system of equations and improve the accuracy of localization, which has a great effect. Next, we intend to structure the magnetic sources to improve the magnetic field gradients to further promote the accuracy of localization.

## Figures and Tables

**Figure 1 sensors-22-06017-f001:**
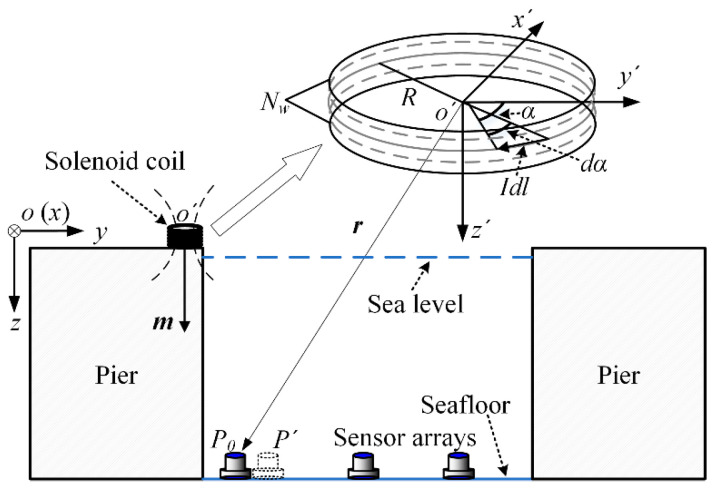
The sensor array arrangement and the solenoid coil setup.

**Figure 2 sensors-22-06017-f002:**
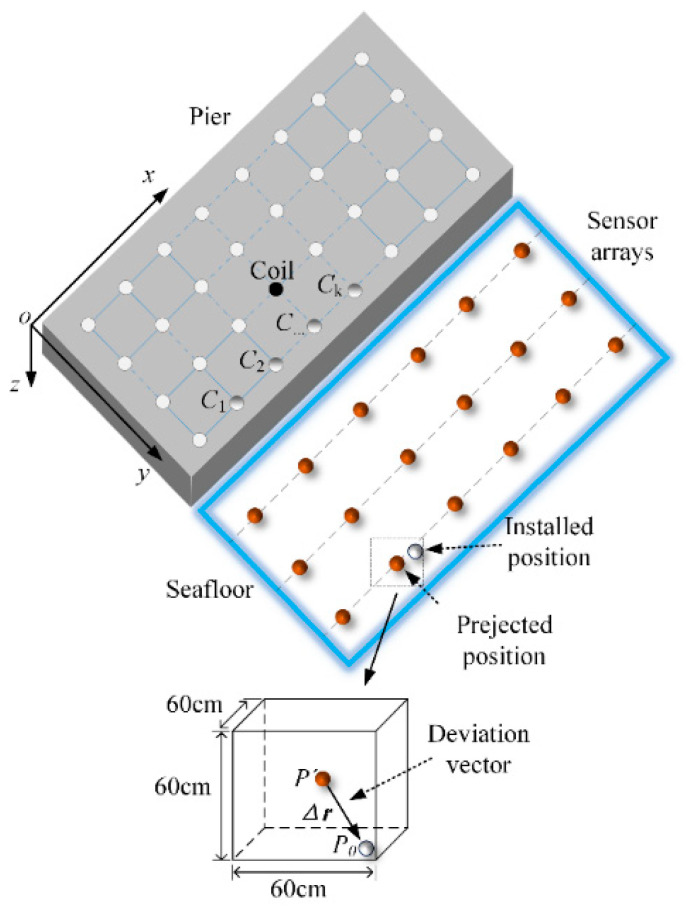
The localization approach for localization by MFG.

**Figure 3 sensors-22-06017-f003:**
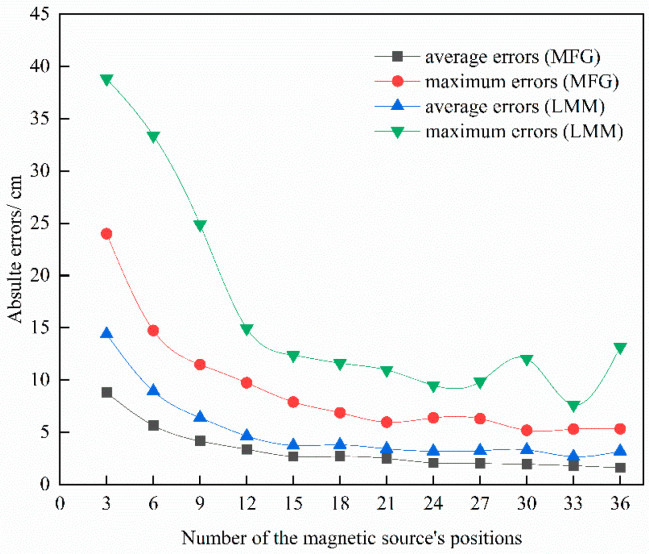
The comparison between LMM and MFG.

**Figure 4 sensors-22-06017-f004:**
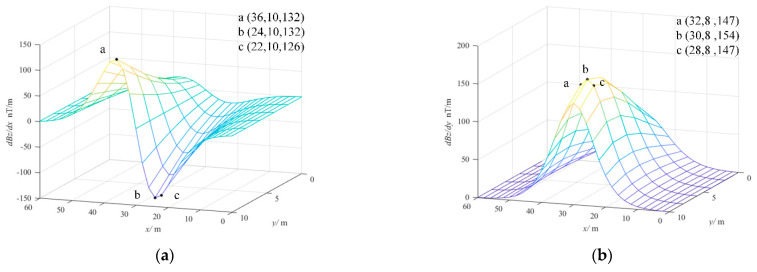
Positions of the magnetic source in the magnetic field and its gradients. (**a**) Positions of the magnetic source in x−direction gradient. (**b**) Positions of the magnetic source in y−direction gradient. (**c**) Positions of the magnetic source in z−direction gradient. (**d**) Positions of the magnetic source in the magnetic field.

**Figure 5 sensors-22-06017-f005:**
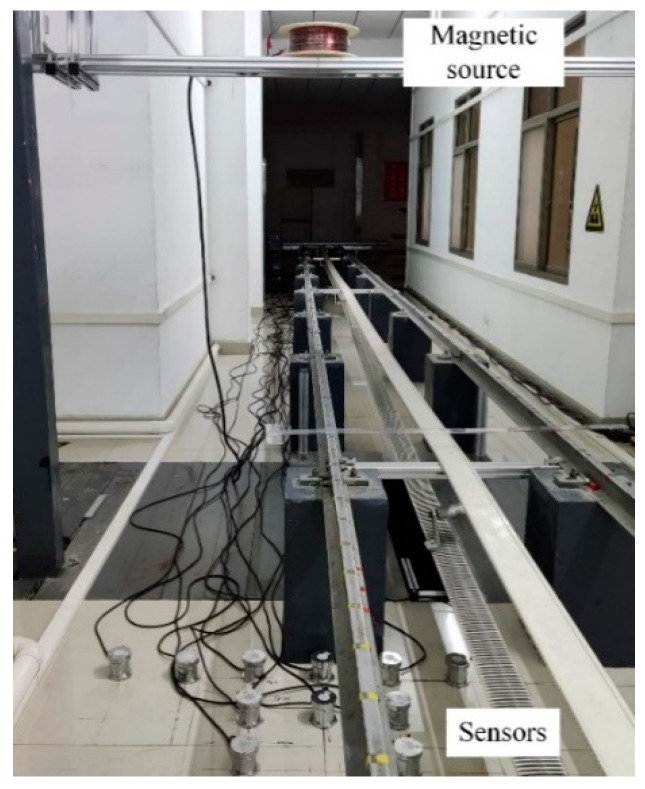
The experimental setup for the localization of the magnetic sensors.

**Table 1 sensors-22-06017-t001:** Average absolute errors with different numbers.

Type	Grid Spacing	Average Absolute Errors with Different Numbers/cm											
		3	6	9	12	15	18	21	24	27	30	33	36
1	1 m	8.7	5.0	3.6	3.6	3.2	2.8	2.7	2.4	2.2	2.3	2.3	2.1
	2 m	8.8	5.6	4.1	3.3	2.7	2.7	2.5	2.1	2.0	1.9	1.8	1.6
	3 m	6.7	4.4	3.2	2.7	2.4	2.3	2.1	2.0	2.0	1.8	1.9	1.7
	4 m	7.0	3.8	2.8	2.6	2.4	2.1	2.1	2.2	2.2	2.1	2.1	2.2
	5 m	5.3	3.8	2.7	2.5	2.7	2.4	2..4	2.4	2.4	2.4	2.4	2.5
2	1 m	15.9	11.4	9.7	9.1	7.0	7.2	7.2	6.4	5.9	5.3	4.8	4.8
	2 m	14.1	10.6	9.2	6.6	5.9	5.3	4.9	4.7	4.3	4.3	4.1	4.1
	3 m	19.5	12.2	6.9	5.7	5.5	5.2	4.5	4.5	4.2	4.2	4.2	4.1
	4 m	19.4	9.8	6.3	5.4	5.1	5.3	5.1	4.6	4.6	4.6	4.6	4.9
	5 m	22.7	8.2	7.0	5.7	5.5	5.4	5.3	5.6	5.1	5.3	5.3	5.5

**Table 2 sensors-22-06017-t002:** Maximum absolute errors with different numbers.

Type	Grid Spacing	Maximum Absolute Errors with Different Numbers/cm											
		3	6	9	12	15	18	21	24	27	30	33	36
1	1 m	23.8	11.1	10.0	9.8	9.3	7.7	7.6	8.0	9.3	8.6	6.0	6.1
	2 m	24.0	14.7	11.5	9.7	7.9	6.9	6.0	6.4	6.3	5.1	5.3	5.3
	3 m	16.8	10.8	9.7	6.5	6.2	5.5	5.1	6.0	4.3	4.6	5.1	4.9
	4 m	15.9	9.2	8.4	6.2	6.1	5.5	5.2	6.0	5.5	4.2	5.8	6.1
	5 m	11.7	10.0	6.5	6.8	6.0	5.8	6.2	6.2	5.7	5.7	6.3	6.8
2	1 m	57.2	26.7	26.4	23.2	20.0	21.1	20.1	17.6	18.7	14.4	11.8	13.5
	2 m	46.5	24.1	26.6	20.1	15.1	18.1	13.8	13.1	11.4	13.1	10.7	10.8
	3 m	77.9	38.1	19.5	17.5	13.5	13.6	14.0	12.6	10.1	11.2	13.1	9.6
	4 m	81.8	22.2	18.4	13.0	12.8	14.4	11.9	11.2	11.5	11.8	12.8	11.2
	5 m	81.1	19.4	20.3	14.3	12.4	14.3	13.1	11.2	11.6	13.1	12.7	12.1

**Table 3 sensors-22-06017-t003:** Average absolute errors with different numbers.

Type	Deviation Vectors/cm	MFG/cm		LMM/cm	
		*d_r_*-Average	*d_r_*-Max	*d_r_*-Average	*d_r_*-Max
1	(30, 0, 0)	3.1	8.1	4.8	16.2
	(0, 30, 0)	3.4	9.7	4.0	15.0
	(0, 0, 30)	3.5	10.5	4.2	13.5
	(30, −30, 0)	2.9	9.4	5.8	14.7
	(30, 30, 30)	3.3	9.7	4.8	13.4
	Average value	3.2	9.5	4.7	14.6
2	(30, 0, 0)	6.4	16.6	6.0	21.0
	(0, 30, 0)	6.4	21.8	8.9	22.4
	(0, 0, 30)	6.8	19.2	6.7	23.7
	(30, −30, 0)	6.7	19.1	7.6	22.8
	(30, 30, 30)	6.6	20.1	7.2	23.1
	Average value	6.6	19.3	7.2	22.6

**Table 4 sensors-22-06017-t004:** The results of the model experiments with different deviation vectors.

Number	Deviation Vectors/cm	MFG		LMM	
		*d_r_*/cm	*d_e_*/%	*d_r_*/cm	*d_e_*/%
1	(3, 0, 0)	0.62	0.3	0.91	0.5
2	(0, 3, 0)	0.93	0.4	1.13	0.6
3	(0, 0, 3)	0.92	0.4	1.45	0.7
4	(3, −3, 0)	0.86	0.4	1.21	0.6
5	(3, 3, 3)	1.02	0.5	1.32	0.7
Average value	-	0.87	0.4	1.20	0.6

## Data Availability

Not applicable.

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
