# Peer review of "Localization Approach for Underwater Sensors in the Magnetic Silencing Facility Based on Magnetic Field Gradients"

_sensors, 2022, doi:10.3390/s22166017_

Round 1
Reviewer 1 Report
Dear authors,
Below I add my comments that I observed in the manuscript:
The summary should show the numerical metrics of the accuracy achieved.
Verify that references are not the journal format.
Reference is required to justifies the phrase “ The importance of the magnetic field has been proverbial since World War II.” Similarly as “But it hardly works in the sea due to incompatibility with water”.
Use space before () and [].
An introduction explaining the purpose of Fig 1 is required in section 2.
The system of equations (1) and (2) are described directly, but the variables and the purpose of the system of equations are not written. In the same way the Localization Approach is not justified correctly as it is implemented in the system.
The results shown in Fig. 3 and Fig. 4 are not correctly justified.
I believe that it is necessary to have a better study of art and make the corresponding comparisons in this type of work.
The details are brief in Figure 5 of the Experimental setup for the location of the magnetic sensors.
In the conclusions, as in the summary, the numerical metrics obtained or quantifiable improvements against state-of-the-art works are not shown.
Best Regards
Author Response
Dear Reviewers:
Thank you for your comments concerning our manuscripts entitled “Localization Approach for Underwater sensors in the Magnetic Silencing Facility based on Magnetic Field Gradients” (ID: Sensors-1790274). These comments are all valuable and very helpful for revising and improving our paper, as well as the important guiding significance to our researches. We have studied comments carefully and have made correction which we hope meet with approval.
For your suggestions and comments, I revise and reply one by one in the attachments.
Best wish!
Yours sincerely,
Yufen Wang (Corresponding author)
E-mail: 18370469195@qq.com

Reviewer 2 Report
1. The significance and novelty of the proposed work is missing, which makes it less interesting for the authors. Moreover, the problem statement of the proposed work must be provided explicitly highlighting the addressed limitations.
2. The manuscript must be provided with a separate section comprising the closely related state-of-the-art studies, especially from 2022 and a comparison between the studies must be provided.
3. The manuscript comprises of the mathematical formulation, which is well appreciated. However, it is unclear whether the equations provided in the manuscript are taken from the existing literature or proposed by the authors themselves.
4. The simulation parameters must be properly discussed. It should be highlighted why the area of the field is selected as such, sensors depth is taken to be 15m, and magnetic dipole moment is set to be 31415A.m2?
5. In Figure 4, the positions of the magnetic source and its gradients must be properly marked in all three different axes. Moreover, the figures’ quality must be improved, paying special attention to the readability.
6. The results show that the relative errors are calculated to be in decimals, e.g., 0.41. Moreover, the values are given in percentage. Are such small percentage values permissible? Kindly confirm.
7. The future work must be provided in the Conclusion section. Also, the numerical results must be discussed.
8. The manuscript must undergo a thorough proofread and the mistakes must be removed.
Author Response

(The authors gave the same response as above.)

Reviewer 3 Report
1. Most of the references in this paper are somewhat too old.
2. The μw in equation 1 is not explained and the formula is problematic.
3. The LMM method in the test was not introduced adequately.
4. It is better to add a set of methods for a comparison.
5. If possible, it is best to experiment in an actual environment to test how much the method is affected by the environment.
Author Response

(The authors gave the same response as above.)

Reviewer 4 Report
In this paper, authors propose a underwater localization method based on magnetic field gradients (MFG). The results show that the proposed method may achieve up to 10cm positioning error in practical use scenario.
In spite of the writing of this paper, my only concern is that the authors did not talk about which method was applied to solve the optimization problem. Adding a short paragraph in Section 3 would help clarify.
Author Response

(The authors gave the same response as above.)

Round 2
Reviewer 1 Report
Dear authors,
In relation to the comments that I previously mentioned, I consider that the metrics in the abstract and the conclusions do not justify a considerable improvement of this work.
I also consider that there is a lack of a correct comparison with state of the art to create a proper table with metrics and existing methodologies, the contribution is not clear.
Additionally, I see different types of font sizes and format in the article and even on the answer sheet.
Best Regards
Author Response

(The authors gave the same response as above.)

Reviewer 2 Report
Now paper can be published.
Author Response
Dear Reviewers:
Thank you for your receptions of our manuscripts entitled “Localization Approach for Underwater sensors in the Magnetic Silencing Facility based on Magnetic Field Gradients” (ID: Sensors-1790274).
Once again, thank you very much for your comments and suggestions.
Yours sincerely,
Yufen Wang (Corresponding author)
E-mail: 18370469195@qq.com